# Levels and Pattern of Alcohol Consumption among Adolescents in Bolivia: A National Cross-Sectional Survey in 2018

Esther Luwedde [1] and Karl Peltzer [1,2,*]

[1]  Department of Psychology, College of Medical and Health Science, Asia University, Taichung 41354, Taiwan
[2]  Department of Psychology, University of the Free State, Bloemfontein 9300, South Africa
*   Correspondence: kfpeltzer@gmail.com

**Abstract:** The purpose of this investigation was to assess the prevalence and associated factors of four alcohol use indicators among male and female school adolescents in Bolivia. In total, 7931 participants (M = 15.5 years, SD = 1.6) responded to a questionnaire in a cross-sectional nationally representative school survey in Bolivia in 2018. The proportion of current alcohol use was 26.4%, heavy alcohol use 11.1%, ever having been drunk 24.1%, and trouble resulting from alcohol use 21.4%. Among boys, older age, current cannabis use, multiple sexual partners, being in a physical fight, school truancy, soft drink intake, injury and psychological distress increased the risk of current alcohol use and/or heavy drinking. Among girls, older age, multiple sexual partners, fast food intake, being in a physical fight, school truancy, sedentary behaviour and psychological distress increased the risk of current alcohol use and/or heavy drinking. Older age, multiple sexual partners, current cannabis use, low parental support and school truancy were associated with trouble from alcohol use and history of intoxication in both sexes. Among boys, ever having used amphetamines, fast food intake, injury, peer support, and being in a physical fight were associated with ever having been drunk; and among girls, sedentary behaviour and psychological distress increased the odds of ever having been drunk. The study found that more than one in ten adolescents engage in heavy alcohol use, and several sex specific factors are identified for four alcohol use indicators.

**Keywords:** alcohol intake; heavy drinking; school adolescents; Bolivia





## 1. Introduction

Worldwide, alcohol misuse among 20–24-year-olds contributes to 7.0% of disability-adjusted life-years [1]. According to Nixon and McClain [2] the adolescence period is marked by being vulnerable to the initiation of alcohol use, which can persist into adulthood. Alcohol consumption at a young age has various adverse health consequences, including violence and injuries, suicidal behaviour, and communicable and noncommunicable illnesses [3,4]. Among adolescents, the prevalences of history of intoxication and current alcohol use were 17.9% and 25.0%, respectively, in low- and middle-income countries (LMICs) [5]. The prevalence of current alcohol consumption among school adolescents was 34.1% in Chile and 59.5% in Uruguay [6], and 45.5% in La Paz, Bolivia, in 2004 [7]. According to estimates of the World Health Organization (2018), 16.6% of adolescents (15–19 years) in Bolivia engage in heavy episodic drinking (≥60 g pure alcohol/past 30 days). In school-based surveys of adolescents in Peru in 2007, and Bolivia in 2012, the prevalence of alcohol use indicators were in terms of current alcohol use in Peru (boys: 32.4%; girls: 27.8%) and in Bolivia (boys: 20.5%; girls 17.7%), regarding history of intoxication in Peru (boys: 20.2%; girls: 12.3%) and in Bolivia (boys: 17.7%; girls: 11.0%), and in terms of trouble from alcohol in Peru (boys: 16.9%; girls: 11.1%) and Bolivia (boys: 14.2%; girls: 10.8%) [8]. More recent national data on the level and pattern of alcohol use among adolescents in Bolivia are lacking, which led to this study.

Previous investigations [9–11] showed that sociodemographic factors (age and sex), mental health factors (psychological distress), health risk behaviours (other substance

use, sedentary behaviour), interpersonal factors (violence and injury), school-level factors (truancy, peer support and bullying victimisation), and parental support factors are associated with alcohol consumption level and pattern. However, we lack information on the multilevel factors associated with alcohol use indicators in Bolivia. The purpose of the current investigation was to assess the prevalence and associated factors of current alcohol use, heavy drinking, history of intoxication and trouble from alcohol use among in-school adolescents in Bolivia in 2018.

## 2. Methods

This research involved an analysis of cross-sectional school health data from a survey conducted in 2018 in Bolivia. Using a two-stage cluster sampling strategy, a nationally representative sample was produced of all students in the 2nd Secondary–6th Secondary in Bolivia; the overall (school and student) response rate was 79% [12]. In the initial stage, schools were chosen on the basis of probability proportional to enrolment size, and at the latter stage, random selection was applied to the classes, and students who were eligible to participate were selected [12]. Approval was granted by Bolivia's Ministry of Health, along with a national ethics committee, to proceed with the study protocol, and written informed consent was obtained from participants (students and parents).

### 2.1. Measures

The questionnaire used in this paper is shown in Table 1. The four outcome variables included current alcohol use, heavy alcohol use ($\geq$2 drinks/day), lifetime drunkenness, and lifetime trouble from alcohol use. Psychological distress was measured with two items by dichotomizing the total score between loneliness and anxiety ($\geq$3 scores, 0–6 range) [13].

**Table 1.** Variable description.

| Variables | Question | Response Options (Coding Scheme) |
|---|---|---|
| Age | "How old are you?" | 11 years old or younger to 18 years old or older |
| Sex | "What is your sex?" | Male, Female |
| Hunger | "During the past 30 days, how often did you go hungry because there was not enough food in your home?" | 1 = never to 5 = always (coded 1–3 = 0 and 4–5 = 1) |
| Current alcohol use | "During the past 30 days, on how many days did you have at least one drink containing alcohol? (This includes drinking beer, wine, singani, rum, vodka, whisky, chichi, or guarapo. Drinking alcohol does not include drinking a few sips of wine for religious purposes. A 'drink' is a glass of wine, a bottle of beer, a small glass of liquor, or a mixed drink)." | 1 = 0 days to 7 = All 30 days (coded 1 = 0 and 2–7 = 1) |
| Heavy alcohol use ($\geq$2 drinks/day) | "During the past 30 days, on the days you drink alcohol, how many drinks did you usually drink per day?" | 1 = Did not drink in the past 30 days to 7 = 5 or more drinks (coded 1–3 = 0 and 4–7 = 1) |
| Ever drunk | "During your life, how many times did you drink so much alcohol that you were really drunk?" | 1 = 0 times to 4 = 10 or more times (coded 1 = 0 and 2–4 = 1) |
| Trouble from alcohol use | "During your life, how many times have you got into trouble with your family or friends, missed school, or got into fights, as a result of drinking alcohol?" | 1 = 0 times to 4 = 10 or more times (coded 1 = 0 and 2–4 = 1) |
| Sources of alcohol | "During the past 30 days, how did you usually get the alcohol you drank?" | 1= I did not drink alcohol during the past 30 days to 8 = I got it some other way |
| Cannabis use | "During the past 30 days, how many times have you used marijuana (also called maria juana, yerba, or mota)? " | 1 = 0 times to 5 = 20 or more times (coded 1 = 0 and 2–5 = 1) |

**Table 1.** *Cont.*

| Variables | Question | Response Options (Coding Scheme) |
|---|---|---|
| Amphetamine use | "During your life, how many times have you used amphetamines or methamphetamines (also called pastillas or pepas)?" | 1 = 0 times to 5 = 20 or more times (coded 1 = 0 and 2–5 = 1) |
| Multiple sexual partners | "During your life, with how many people have you had sexual intercourse?" | 1 = I have never had sexual intercourse to 7 = 6 or more people (coded 0–1 = 0 and 2–7 = 1) |
| Soft drink consumption | "During the past 7 days, how many times did you drink a can, bottle, or glass of a carbonated soft drink, such as Pepsi, Coca-Cola, Fanta, Papaya Salvieti or Kinoto? (Do not include diet soft drinks.)" | 1 = not in the past 7 days to 7 = 5 or more times per day (coded 1–2 = 0 and 3–7 = 1) |
| Fast food consumption | "During the past 7 days, on how many days did you eat food from a fast food restaurant, such as Burger King, Pollos Copacabana, Dumbo, Toby's or Pizza Elli's?" | 0–7 days (coded 0–1 day = 0 and 3–7 days = 1) |
| School truancy | "During the past 30 days, on how many days did you miss classes or school without permission?" | 1 = 0 days to 5 = 10 or more days (coded 1 = 0 and 2–5 = 1) |
| In a physical fight | "During the past 12 months, how many times were you in a physical fight?" | 1 = 0 times to 8 = 12 or more times (coded 1 = 0 and 2–8 = 1) |
| Physically attacked | "During the past 12 months, how many times were you physically attacked?" | 1 = 0 times to 8 = 12 or more times (coded 1 = 0 and 2–8 = 1) |
| Injury | "During the past 12 months, how many times were you seriously injured?" | 1 = 0 times to 8 = 12 or more times (coded 1 = 0 and 2–8 = 1) |
| Bullied in school | "During the past 12 months, have you ever been bullied on school property?" | Yes/No (coded 0 = no, 1 = yes) |
| Bullied outside school | "During the past 12 months, have you ever been bullied when you were not on school property?" | Yes/No (coded 0 = no, 1 = yes) |
| Cyber bullied | "During the past 12 months, have you ever been cyber bullied?" | Yes/No (coded 0 = no, 1 = yes) |
| Sedentary behaviour | "How much time do you spend during a typical or usual day sitting and watching television, playing computer games, talking with friends, or doing other sitting activities, such as playing with the mobile phone or videogames (when you are not at school or doing homework)?" | 1 =< 1 h/day to 6 => 8 h/day (coded < 3 h/day = 0 and ≥3 h/day = 1) |
| Psychological distress | | |
| Anxiety | "During the past 12 months, how often have you been so worried about something that you could not sleep at night?" | 1 = never to 5 = always |
| Loneliness | "During the past 12 months, how often have you felt lonely?" | 1 = never to 5 = always |
| Protective factors | | |
| Peer support | "During the past 30 days, how often were most of the students in your school kind and helpful?" | 1 = never to 5 = always (coded 1–3 = 0 and 4–5 = 1) |
| Parental supervision | "During the past 30 days, how often did your parents or guardians check to see if your homework was done?" | 1 = never to 5 = always (coded 1–3 = 0 and 4–5 = 1) |
| Parental connectedness | "During the past 30 days, how often did your parents or guardians understand your problems and worries?" | 1 = never to 5 = always (coded 1–3 = 0 and 4–5 = 1) |
| Parental bonding | "During the past 30 days, how often did your parents or guardians really know what you were doing with your free time? | 1 = never to 5 = always (coded 1–3 = 0 and 4–5 = 1) |
| Parental respect for privacy | "During the past 30 days, how often did your parents or guardians go through your things without your approval?" | 1 = never to 5 = always (coded 1–3 = 0 and 4–5 = 1) |

The four parental support items (0–4) were summed up and re-categorised into low (0–1), medium (2) and high (3–4) [10].

*2.2. Statistical Analysis*

Frequency calculations were used to describe the sample, and differences in proportions were tested using chi-square statistics. Adjusted prevalence ratios were estimated for the four alcohol use indications using Poisson regression analyses. $p < 0.05$ was considered significant, and missing data were discarded. All analyses were conducted using STATA software version 15.0 (Stata Corporation, College Station, TX, USA), taking the multistage sampling and weighting of the data into account.

## 3. Results

The sample included 7931 school students (mean age 15.5 years, 1.6 SD) from Bolivia. Table 1 shows the basic sample characteristics and alcohol use indicators. The prevalence of current alcohol use was 26.4%, heavy alcohol use 11.1%, ever having been drunk 24.1%, and trouble from alcohol use 21.4%. Current alcohol, heavy drinking and trouble from alcohol use did not significantly differ by sex; only lifetime drunkenness was significantly higher among boys than girls. Almost one in ten (7.8%) of the students were current cannabis users, and 4.5% had ever used amphetamine. More than half (52.1%) of participants reported a serious injury (past 12 months), 40.8% had engaged in school truancy (past month), 33.7% consumed one or more soft drinks per day, 32.9% had fast food on two or more days (past week), 31.6% engaged in sedentary behaviour, and 30.6% had participated in physical fighting (past 12 months). In terms of sourcing alcohol, 46.6% obtained it from family or friends and 37.6% had directly bought alcoholic beverages from the shop (see Table 2).

**Table 2.** Descriptive results, school adolescents, Bolivia, 2018.

| Variable | Sample | Current Alcohol | Two or More Drinks/Day | Ever Drunk | Trouble from Alcohol Use |
|---|---|---|---|---|---|
| | N (%) | % | % | % | % |
| All | 7931 | 26.4 | 11.1 | 24.1 | 21.4 |
| Age in years | | | | | |
| ≤14 | 2366 (30.5) | 18.4 | 6.2 | 13.7 | 15.1 |
| 15 | 1593 (20.4) | 23.2 | 9.7 | 18.8 | 20.5 |
| 16 | 1591 (19.9) | 25.9 | 11.6 | 25.2 | 21.6 |
| ≥17 | 2286 (29.3) | 36.7 | 16.6 | 37.8 | 28.0 |
| Sex | | | | | |
| Female | 3763 (49.3) | 24.6 | 10.1 | 21.0 | 20.6 |
| Male | 4041 (50.7) | 27.5 | 11.7 | 26.6 * | 21.5 |
| Hunger | 317 (4.1) | 33.2 | 16.0 | 31.2 | 32.1 |
| Current cannabis use | 574 (7.8) | 67.4 | 37.1 | 67.3 | 62.1 |
| Ever amphetamine use | 312 (4.5) | 67.7 | 31.1 | 69.5 | 70.5 |
| Sex partners (≥two) | 949 (13.1) | 54.7 | 31.5 | 57.7 | 38.7 |
| Soft drink consumption | 2625 (33.7) | 29.2 | 12.6 | 26.0 | 24.9 |
| Fast food consumption | 2623 (32.9) | 34.5 | 15.8 | 30.4 | 26.8 |
| School truancy | 3112 (40.8) | 36.4 | 16.2 | 34.9 | 32.2 |
| Bullied in school | 1824 (24.7) | 27.9 | 11.0 | 25.2 | 26.1 |
| Bullied outside school | 1641 (21.7) | 32.8 | 12.8 | 29.0 | 32.2 |
| Cyberbullied | 1615 (21.4) | 34.1 | 14.2 | 31.6 | 30.8 |
| In physical fight | 2370 (30.6) | 42.1 | 18.3 | 37.3 | 35.0 |
| Physically attacked | 2208 (28.6) | 36.2 | 15.0 | 33.5 | 34.4 |

**Table 2.** *Cont.*

| Variable | Sample | Current Alcohol | Two or More Drinks/Day | Ever Drunk | Trouble from Alcohol Use |
|---|---|---|---|---|---|
| Injury | 3738 (52.1) | 31.9 | 13.2 | 29.1 | 26.8 |
| Sedentary behaviour | 2356 (31.6) | 32.6 | 16.5 | 29.8 | 26.2 |
| Psychological distress | 1634 (22.3) | 35.8 | 16.1 | 33.1 | 30.8 |
| Peer support | 2617 (34.4) | 26.3 | 12.1 | 23.7 | 18.2 |
| Parental support | | | | | |
| 0–1 | 3659 (51.6) | 30.2 | 11.9 | 28.8 | 27.9 |
| 2 | 1595 (22.0) | 26.0 | 12.3 | 22.6 | 17.9 |
| 3–4 | 1896 (26.4) | 17.5 | 7.6 | 15.0 | 9.9 |

Percentages are weighted. * Significantly higher in males than in females ($p < 0.05$).

### 3.1. Adjusted Associations with Current Alcohol Use and Heavy Drinking

Among both boys and girls, older age, participation in physical fighting, school truancy, and psychological distress increased the odds of current alcohol consumption. In addition, current cannabis use, multiple sexual partners, and injury in the past 12 months, among boys, and fast food intake among girls, increased the odds of current alcohol consumption.

In both sexes, multiple sexual partners increased the risk of heavy alcohol use. In addition, among boys, older age, soft drink intake, current cannabis use and being in a physical fight were associated with heavy drinking, and among girls, school truancy and sedentary behaviour were associated with heavy drinking (see Table 3).

**Table 3.** Associations with current alcohol use and heavy alcohol use by sex.

| Variable | Current Alcohol Use | | Alcohol 2+ Per Day | |
|---|---|---|---|---|
| | **Male** | **Female** | **Male** | **Female** |
| | APR (95% CI) | APR (95% CI) | APR (95% CI) | APR (95% CI) |
| Age (years) | 1.20 (1.11 to 1.29) *** | 1.11 (1.03 to 1.21) * | 1.19 (1.07 to 1.33) ** | 1.10 (0.97 to 1.24) |
| Hunger | 0.74 (0.53 to 1.10) | 1.17 (0.79 to 1.72) | 0.85 (0.37 to 1.93) | 0.73 (0.36 to 1.46) |
| Current cannabis use | 1.44 (1.10 to 1.87) ** | 1.02 (0.61 to 1.70) | 2.21 (1.56 to 3.11) *** | 1.58 (0.79 to 3.13) |
| Ever amphetamine use | 1.19 (0.83 to 1.72) | 1.25 (0.68 to 2.27) | 1.28 (0.86 to 1.91) | 1.25 (0.51 to 3.09) |
| Sex partners (≥two) | 1.61 (1.33 to 1.94) *** | 1.68 (1.34 to 2.11) | 2.13 (1.57 to 2.87) *** | 2.60 (1.65 to 4.10) *** |
| Soft drink consumption | 1.04 (0.86 to 1.25) | 1.18 (0.99 to 1.42) | 1.26 (1.07 to 1.48) *** | 1.13 (0.36 to 1.46) |
| Fast food consumption | 1.23 (0.98 to 1.52) | 1.31 (1.04 to 1.65) * | 1.28 (0.88 to 1.87) | 1.22 (0.87 to 1.72) |
| School truancy | 1.23 (1.08 to 1.39) ** | 1.26 (1.05 to 1.51) * | 1.00 (0.65 to 1.53) | 1.51 (1.06 to 2.16) * |
| Bullied | | | | |
| No | 1 (Reference) | 1 (Reference) | 1 (Reference) | 1 (Reference) |
| 1 type | 0.96 (0.74 to 1.25) | 1.04 (0.80 to 1.35) | 0.94 (0.61 to 1.44) | 0.94 (0.71 to 1.26) |
| 2–3 types | 1.10 (0.86 to 1.42) | 1.09 (0.83 to 1.43) | 1.40 (0.89 to 2.20) | 0.90 (0.56 to 1.46) |
| In physical fight | 1.75 (1.35 to 1.66) *** | 1.49 (1.17 to 1.91) ** | 2.01 (1.37 to 2.93) *** | 1.32 (0.89 to 1.97) |
| Physically attacked | 1.08 (0.91 to 1.30) | 0.97 (0.78 to 1.20) | 0.88 (0.60 to 1.30) | 1.05 (0.74 to 1.49) |
| Injury | 1.31 (1.03 to 1.66) * | 1.24 (0.97 to 1.60) | 1.36 (0.89 to 2.07) | 1.36 (0.89 to 2.07) |
| Sedentary behaviour | 1.06 (0.81 to 1.38) | 1.19 (0.98 to 1.45) | 1.34 (0.91 to 1.98) | 1.99 (1.56 to 2.53) *** |
| Psychological distress | 1.36 (1.13 to 1.65) ** | 1.40 (1.14 to 1.70) ** | 1.37 (0.95 to 1.99) | 1.40 (0.90 to 2.16) |
| Peer support | 1.11 (0.87 to 1.41) | 1.22 (0.99 to 1.52) | 1.13 (0.77 to 1.66) | 1.34 (0.96 to 1.88) |
| Parental support | | | | |
| 0–1 | 1 (Reference) | 1 (Reference) | 1 (Reference) | 1 (Reference) |
| 2 | 0.65 (0.48 to 0.87) ** | 1.09 (0.85 to 1.39) | 0.80 (0.49 to 1.30) | 1.14 (0.87 to 1.49) |
| 3–4 | 0.80 (0.60 to 1.07) | 0.86 (0.64 to 1.16) | 0.87 (0.56 to 1.37) | 0.75 (0.47 to 1.20) |

CPR = Crude Prevalence Ratio; CI = Confidence Interval; *** $p < 0.001$; ** $p < 0.01$; * $p < 0.05$.

### 3.2. Adjusted Associations with History of Intoxication and Trouble from Alcohol Use

In both sexes, older age, current cannabis use, multiple sexual partners, school truancy, and low parental support were associated with history of intoxication and alcohol-related problems. Among boys, ever having used amphetamines, fast food intake, participation in physical fighting, peer support and injury were associated with ever having been drunk, and among girls, sedentary behaviour and psychological distress were associated with ever having been drunk. Furthermore, among boys, injury, and among girls, psychological distress, being bullied, soft drink intake and history of amphetamine use were associated with alcohol related problems (see Table 4).

**Table 4.** Associations with lifetime drunkenness and trouble from alcohol use by sex.

| Variable | Ever Drunk | | Trouble from Alcohol Use | |
|---|---|---|---|---|
| | Male | Female | Male | Female |
| | APR (95% CI) | APR (95% CI) | APR (95% CI) | APR (95% CI) |
| Age (years) | 1.35 (1.24 to 1.47) *** | 1.13 (1.05 to 1.22) *** | 1.11 (1.02 to 1.20) * | 1.13 (1.04 to 1.22) ** |
| Hunger | 1.02 (0.72 to 1.45) | 0.80 (0.48 to 1.35) | 0.86 (0.51 to 1.46) | 0.79 (0.49 to 1.27) |
| Current cannabis use | 1.40 (1.12 to 1.74) ** | 2.00 (1.45 to 2.75) *** | 1.63 (1.23 to 2.17) *** | 1.55 (1.12 to 2.16) ** |
| Ever amphetamine use | 1.45 (1.00 to 1.54) * | 1.57 (0.95 to 2.59) | 1.21 (0.83 to 1.77) | 1.70 (1.16 to 2.50) ** |
| Sex partners (≥two) | 1.53 (1.23 to 1.91) *** | 2.04 (1.54 to 2.70) *** | 1.44 (1.15 to 1.81) ** | 1.44 (1.02 to 1.58) * |
| Soft drink consumption | 0.93 (0.74 to 1.16) | 1.02 (0.80 to 1.31) | 0.95 (0.78 to 1.16) | 1.27 (1.02 to 1.58) * |
| Fast food consumption | 1.24 (1.01 to 1.53) * | 1.06 (0.85 to 1.33) | 1.21 (0.97 to 1.51) | 0.99 (0.80 to 1.23) |
| School truancy | 1.48 (1.21 to 1.81) *** | 1.54 (1.29 to 1.84) *** | 1.61 (1.28 to 2.03) *** | 1.88 (1.52 to 2.33) *** |
| Bullied<br>No<br>1 type<br>2–3 types | <br>1 (Reference)<br>0.98 (0.79 to 1.20)<br>1.13 (0.82 to 1.56) | <br>1 (Reference)<br>0.94 (0.68 to 1.30)<br>0.68 (0.49 to 0.96) * | <br>1 (Reference)<br>1.14 (0.91 to 1.42)<br>1.20 (0.95 to 1.50) | <br>1 (Reference)<br>1.34 (1.00 to 1.80) *<br>1.16 (0.88 to 1.52) |
| In physical fight | 1.41 (1.11 to 1.78) ** | 1.16 (0.85 to 1.59) | 1.16 (0.88 to 1.52) | 1.23 (0.97 to 1.55) |
| Physically attacked | 0.95 (0.75 to 1.20) | 1.06 (0.86 to 1.31) | 1.23 (0.95 to 1.61) | 1.09 (0.90 to 1.33) |
| Injury | 1.24 (1.00 to 2.09) * | 1.23 (0.99 to 1.53) | 1.63 (1.16 to 2.28) ** | 1.26 (0.94 to 1.68) |
| Sedentary behaviour | 1.12 (0.90 to 1.39) | 1.29 (1.04 to 1.60) * | 1.24 (0.96 to 1.61) | 0.96 (0.76 to 1.20) |
| Psychological distress | 1.05 (0.85 to 1.31) | 1.40 (1.15 to 1.70) *** | 1.10 (0.86 to 1.40) | 1.28 (1.02 to 1.61) * |
| Peer support | 1.36 (1.06 to 1.73) * | 1.32 (1.03 to 1.69) | 1.15 (0.94 to 1.41) | 1.20 (0.94 to 1.54) |
| Parental support<br>0–1<br>2<br>3–4 | <br>1 (Reference)<br>0.77 (0.57 to 1.03)<br>0.73 (0.55 to 0.97) * | <br>1 (Reference)<br>0.87 (0.70 to 1.09)<br>0.63 (0.46 to 0.87) ** | <br>1 (Reference)<br>0.55 (0.40 to 0.77) ***<br>0.66 (0.50 to 0.86) ** | <br>1 (Reference)<br>0.68 (0.51 to 0.91) *<br>0.52 (0.36 to 0.74) *** |

APR = Adjusted Prevalence Ratio; CI = Confidence Interval; *** $p < 0.001$; ** $p < 0.01$; * $p < 0.05$.

## 4. Discussion

The purpose of the study was to assess the prevalence and associated factors of current alcohol consumption, heavy alcohol use, history of intoxication and trouble from alcohol use among adolescents in a nationally representative school survey in Bolivia in 2018. We found a prevalence of current alcohol consumption of 26.4% (27.5% in boys; 24.6% in girls), which is higher than in 57 LMICs (25.0%) [5] and higher than in the 2012 Bolivia GSHS (20.5% in boys; 17.7% in girls) [8], but lower than in Peru (32.4% in boys; 27.8% in girls) [8], in Chile (34.1%) and in Uruguay (59.5%) [6], and in La Paz, Bolivia, in 2004 (45.5%) [7]. The found prevalence of heavy alcohol use, at 11.1%, was lower than World Health Organization [4] estimates of 16.6% (in adolescents, 15–19 years) in Bolivia. History of intoxication was 24.1% prevalent (26.6% among boys and 21.0% among girls), which is

higher than in 57 LMICs (17.9%) [5], higher than in the 2012 Bolivia GSHS (17.7% among boys and 11.0% among girls), and higher than in Peru (20.2% among boys and 12.3% among girls) [8]. The prevalence of trouble from alcohol use of 21.4% (21.5% among boys and 20.6% among girls) was higher than in the 2012 Bolivia GSHS (14.2% among boys and 10.8% among girls), and in Peru (16.9% among boys and 11.1% among girls) [8].

On three different alcohol use indicators (current alcohol use, history of intoxication and trouble from alcohol use), an increase from the 2012 to the 2018 GSHS in Bolivia was observed; this may be attributed to the nonexistence of alcohol policy control measures, such as:

"no written national policy, no national action plan, no excise tax on beer/wine/spirits, no national legal minimum age for off-premise and on-premise sales of alcoholic beverages (beer/wine/spirits), no legally binding regulations on alcohol advertising, and no legally required health warning labels on alcohol advertisements/containers," [4].

For example, an increase in alcohol marketing in Bolivia may have contributed to an increase of alcohol consumption among adolescents in Bolivia [14]. Furthermore, we found that a large proportion of current alcohol users (37.6%) sourced their alcohol from a shop, and this was found to be significantly associated with heavy drinking [15].

The prevalences of current alcohol use, history of intoxication and trouble from alcohol use in this study were not significantly higher among boys than girls, unlike a large multicountry study among school adolescents that found males had higher odds of current alcohol consumption, trouble from alcohol use, and lifetime drunkenness [8]. However, boys had a significantly higher prevalence of heavy drinking than girls in this study. The prevalences of the four alcohol use indicators significantly increased with age in this study, which is in agreement with previous findings [5].

In terms of mental health factors, this survey showed that psychological distress was associated with current alcohol use, consistent with previous studies [16,17]. Among girls, psychological distress was also associated with history of intoxication and trouble from alcohol use. It is possible that adolescents engage in alcohol use or misuse as a way of coping with psychological distress [17]. Regarding health risk behaviours, current cannabis use, history of amphetamine use, fast food consumption, sedentary behaviour and multiple sexual partners, were found to be associated among boys and/or girls with one or more alcohol use indicators. These findings are largely consistent with earlier studies [10,18–20].

Regarding interpersonal factors, physical fighting and injury were, particularly among boys, associated with several alcohol use indicators. This is in line with previous research [10]. Of the school level factors, school truancy was consistently associated across alcohol use indicators, in agreement with former investigations [18], while bullying victimisation and lack of peer support at school were not associated with alcohol use indicators, except that among boys, peer support at school was positively associated with history of intoxication and among girls, exposure to one type of bullying victimisation was associated with trouble from alcohol use. It is possible that participants are truant from school to engage in alcohol use and misuse, and that boys drink alcohol among their peers who encourage them to get drunk. This result emphasises the need to improve school attendance by increasing social cohesion in school [18].

Regarding parental support factors, we found that, consistent with some previous studies [7,21], lack of parental support was associated among both sexes with history of intoxication and trouble from alcohol use. Overall, the study found possible clustering of mental health, health risk behaviour, interpersonal, school, and parental factors associated with alcohol use indicators; these factors should be incorporated into multilevel school health promotion interventions targeting adolescents to reduce alcohol misuse [10,18,22].

*Study Limitations*

Data were assessed by self-report and in a cross-sectional school survey, which may have produced biased responses, and data cannot be generalised to the adolescent population in Bolivia. Out-of-school adolescents may have higher rates of alcohol use and

misuse than in-school adolescents. Some variables such as tobacco use and parental and peer substance use, which may have influenced alcohol use, were not measured in this survey and should form part of future investigations.

## 5. Conclusions

One in four adolescent students were current alcohol users and had a history of intoxication. Among girls and/or boys, multiple risk behaviours (cannabis use, sexual risk behaviour, amphetamine use, fast food and soft drink intake, physical fighting, sedentary behaviour and school truancy), lack of parental support and psychological distress were associated with current alcohol consumption, heavy drinking, history of intoxication and/or trouble from alcohol use. Considering the multiple risk factors, health promotion is suggested to reduce misuse of alcohol among adolescents in Bolivia.

**Author Contributions:** Conceptualization, E.L.; Formal analysis, K.P.; Methodology, E.L.; Validation, K.P.; Writing—original draft, E./L.; Writing—review & editing, K.P. All authors have read and agreed to the published version of the manuscript.

**Funding:** This research received no external funding.

**Institutional Review Board Statement:** Approval was granted by Bolivia's Ministry of Health, along with a national ethics committee, to proceed with the study protocol.

**Informed Consent Statement:** Written informed consent was obtained from participants (students and parents).

**Data Availability Statement:** The data on which this paper is based are available at the World Health Organization NCD Microdata Repository, at https://extranet.who.int/ncdsmicrodata/index.php/catalog/881 (accessed on 5 September 2021).

**Acknowledgments:** This paper uses data from the Global School-Based Student Health Survey (GSHS). GSHS is supported by the World Health Organization and the US Centers for Disease Control and Prevention.

**Conflicts of Interest:** The authors declare no conflict of interest.

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
