# Peer review of "Levels and Pattern of Alcohol Consumption among Adolescents in Bolivia: A National Cross-Sectional Survey in 2018"

_adolescents, doi:10.3390/adolescents3010001_

Round 1

Reviewer 1 Report

This is a good manuscript. I only have a few comments as follows.

- In the introduction, I was wondering why the authors only compared the prevalence of alcohol use between Peru and Bolivia. In addition, this study was based on the most recent GSHS data collected in 2018. I would like to know whether any previous studies have been published on this topic and used an earlier survey round of 2012 GSHS. The authors could cite those studies (if any) and highlight something "new" they added in this present study.

- In the method section, please provide information on how many schools were selected and the response rate of the participants.

- "Psychological distress (≥3 scores, 0-6 range) was measured with two items (Pengpid & Peltzer, 2020)"; please mention explicitly that psychological distress was defined by dichotomising the total score between loneliness and anxiety.

- “During the past 12 months, how often have you felt lonely?” and “During the past 12 months, how often have you been so worried about something that you could not sleep at night?” This sentence can be removed since the readers can look at the table.

-  "The four parental support items (0-4) were grouped into low (0-1), medium (2) and high (3-4) (Pengpid & Peltzer, 2019)." Similar to the psychological distress measure, the authors need to mention that four parental support items were added together/summed up first before being re-categorised. 

- Please mention explicitly whether the authors reported weighted percentages in Table 1.

- "Current alcohol, heavy drinking and trouble from alcohol use did not significantly differ by sex, only lifetime drunkenness was significantly higher among boys than girls. " Please mention what statistical analysis was used to determine this finding in the method section.

- "On the different alcohol use indicators assessed an increase from the 2012 to the 2018 GSHS in Bolivia was observed, which may be attributed to the non-existence of alcohol policy control measures, such as". It would be better if the authors could rewrite this sentence. I also would like to know what the authors meant by "different alcohol use indicators".

Author Response

Reviewer I

Comments and Suggestions for Authors
This is a good manuscript. I only have a few comments as follows.
- In the introduction, I was wondering why the authors only compared the prevalence of alcohol use between Peru and Bolivia. In addition, this study was based on the most recent GSHS data collected in 2018. I would like to know whether any previous studies have been published on this topic and used an earlier survey round of 2012 GSHS. The authors could cite those studies (if any) and highlight something "new" they added in this present study.
Response: this is stated, as in below
Among school-based adolescent surveys in Peru in 2007, and Bolivia in 2012 the prevalence of  alcohol use indicators were in terms of current alcohol use in Peru (boys: 32.4%; girls: 27.8%) and in Bolivia (boys: 20.5%; girls 17.7%), regarding history of intoxication in Peru (boys: 20.2%; girls: 12.3%) and in Bolivia (boys: 17.7%; girls: 11.0%), and terms of trouble from alcohol in Peru (boys: 16.9%; girls: 11.1%) and Bolivia (boys: 14.2%; girls: 10.8%) (Leung et al., 2019). More recent national data on the level and pattern of alcohol use among adolescents in Bolivia are lacking, which led to this study.

- In the method section, please provide information on how many schools were selected and the response rate of the participants.
Response: below is added (number of schools is not known) now in the method section (it was stated under results)
the overall (school and student) response rate was 79%
- "Psychological distress (≥3 scores, 0-6 range) was measured with two items (Pengpid & Peltzer, 2020)"; please mention explicitly that psychological distress was defined by dichotomising the total score between loneliness and anxiety.
Response: added accordingly
- “During the past 12 months, how often have you felt lonely?” and “During the past 12 months, how often have you been so worried about something that you could not sleep at night?” This sentence can be removed since the readers can look at the table.
Response: removed accordingly

-  "The four parental support items (0-4) were grouped into low (0-1), medium (2) and high (3-4) (Pengpid & Peltzer, 2019)." Similar to the psychological distress measure, the authors need to mention that four parental support items were added together/summed up first before being re-categorised. 
Response: added accordingly
- Please mention explicitly whether the authors reported weighted percentages in Table 1.
Response: added
- "Current alcohol, heavy drinking and trouble from alcohol use did not significantly differ by sex, only lifetime drunkenness was significantly higher among boys than girls. " Please mention what statistical analysis was used to determine this finding in the method section.
Response: below is stated in the method section
differences in proportions were tested using chi-square statistics
- "On the different alcohol use indicators assessed an increase from the 2012 to the 2018 GSHS in Bolivia was observed, which may be attributed to the non-existence of alcohol policy control measures, such as". It would be better if the authors could rewrite this sentence.
 I also would like to know what the authors meant by "different alcohol use indicators".
Response: below is added
On the different alcohol use indicators (current alcohol use, history of intoxication and trouble from alcohol use)

Reviewer 2 Report

The study is interesting but represents only the first part of who need to be an accurate analysis from a scientific point of view. the data presented are purely descriptive and do not provide a clear and in-depth analysis of the situation.

The fact that statistical associations exist does not determine a cause and effect relationship. Therefore, I suggest expanding the analyzes by investigating the possible predictors of alcohol consumption among the variables investigated and for which an association has been highlighted.

Moreover the discussion it isn't a report of the results but must be an argumention of the results in light to the scientific  literature. Reporting all the percentages obtained in the discussions also makes it difficult to read and understand what must be argued instead.

Author Response

Reviewer 2

Comments and Suggestions for Authors
The study is interesting but represents only the first part of who need to be an accurate analysis from a scientific point of view. the data presented are purely descriptive and do not provide a clear and in-depth analysis of the situation.
Response: disagree, as we provide 4 different logistic regression models with 4 alcohol use indicators
The fact that statistical associations exist does not determine a cause and effect relationship. Therefore, I suggest expanding the analyzes by investigating the possible predictors of alcohol consumption among the variables investigated and for which an association has been highlighted.
Response: we report under study limitations that 
causality cannot be established
Moreover the discussion it isn't a report of the results but must be an argumention of the results in light to the scientific  literature. 
Response: this is what has been done
Reporting all the percentages obtained in the discussions also makes it difficult to read and understand what must be argued instead.
Response: some percentages are needed to compare

Round 2

Reviewer 2 Report

Dear authors 

when a referee underlines a lack, it means that probabily it isn't clearly write and defined in the paper, and readers have difficults to understand  what authors really think to disseminate.

So, I suggest to reconsider what I underlined and try to re-write partially  some sentences of the discussion and limitation sections.

Author Response

Comments and Suggestions for Authors
Dear authors 
when a referee underlines a lack, it means that probabily it isn't clearly write and defined in the paper, and readers have difficults to understand  what authors really think to disseminate.
So, I suggest to reconsider what I underlined and try to re-write partially  some sentences of the discussion and limitation sections.
Response: Accordingly, some changes and additions were made in the discussion and limitation sections.